# Potential for Misinterpretation in the Laboratory Diagnosis of *Clostridioides difficile* Infections

**DOI:** 10.3390/diagnostics15091166

**Published:** 2025-05-03

**Authors:** Alexandra Kalacheva, Metodi Popov, Valeri Velev, Rositsa Stoyanova, Yordanka Mitova-Mineva, Tsvetelina Velikova, Maria Pavlova

**Affiliations:** 1“Ajibadem City Clinic-Tokuda” University General Hospital, 1784 Sofia, Bulgaria; alexandra.kalacheva20@gmail.com; 2“CIBALAB” Medical Diagnostic Laboratory, 1303 Sofia, Bulgaria; 3General Hospital “St. Ivan Rilski”, 1233 Dupnica, Bulgaria; metodi_b_popov@abv.bg; 4Department of Epidemiology and Hygiene, Medical University of Sofia, 1431 Sofia, Bulgaria; 5National Center of Infectious and Parasitic Diseases, Department of Microbiology, The National Reference Laboratory for Enteric Diseases, 1504 Sofia, Bulgariamimipavlovaa@gmail.com (M.P.); 6Medical Faculty, Sofia University St. Kliment Ohridski, 1 Kozyak Str., 1407 Sofia, Bulgaria; tsvelikova@medfac.mu-sofia.bg

**Keywords:** *Clostridioides difficile*, cdtA, cdtB, CDI, laboratory diagnostic

## Abstract

**Background/Objective**. Toxin-producing strains of *Clostridioides difficile* (*C. diff*) are the most commonly identified cause of healthcare-associated infection in the elderly. Risk factors include advanced age, hospitalization, prior or concomitant systemic antibacterial therapy, chemotherapy, and gastrointestinal surgery. Patients with unspecified and new-onset diarrhea with ≥3 unformed stools in 24 h are the target population for *C. diff* infection (CDI) testing. To present data on the risks of laboratory misdiagnosis in managing CDI. **Materials**. In two general hospitals, we examined 116 clinical stool specimens from hospitalized patients with acute diarrhea suspected of nosocomial or antibiotic-associated diarrhea (AAD) due to *C. diff.* Enzyme immunoassay (EIA) tests for the detection of *C. diff* toxins A (cdtA) and B (cdtB) in stool, automated CLIA assay for the detection of *C. diff* GDH antigen and qualitative determination of cdtA and B in human feces and anaerobic stool culture were applied for CDI laboratory diagnosis. MALDI-TOF (Bruker) was used to identify the presumptive anaerobic bacterial colonies. The following methods were used as confirmatory diagnostics: the LAMP method for the detection of Salmonella spp. and simultaneous detection of *C. jejuni* and *C. coli*, an *E. coli* Typing RT-PCR detection kit (ETEC, EHEC, STEC, EPEC, and EIEC), API 20E and aerobic stool culture methods. **Results**. A total of 40 toxigenic strains of *C. diff* were isolated from all 116 tested diarrheal stool samples, of which 38/40 produced toxin B and 2/40 strains were positive for both cdtA and cdtB. Of the stool samples positive for cdtA (6/50) and/or cdtB (44/50) by EIA, 33 were negative for *C. diff* culture but positive for the following diarrheal agents: *Salmonella enterica* subsp. *arizonae* (1/33, LAMP, culture, API 20E); *C. jejuni* (2/33, LAMP, culture, MALDI TOF); ETEC O142 (1/33), STEC O145 and O138 (2/33, *E. coli* RT-PCR detection kit, culture); *C. perfringens* (2/33, anaerobic culture, MALDI TOF); hypermycotic enterotoxigenic *K. pneumonia* (2/33) and enterotoxigenic *P. mirabilis* (2/33, culture; PCR encoding LT-toxin). Two of the sixty-six cdtB-positive samples (2/66) showed a similar misdiagnosis when analyzed using the CLIA method. However, the PCR analysis showed that they were cdtB-negative. In contrast, the LAMP method identified a positive result for *C. jejuni* in one sample, and another was STEC positive (stx1+/stx2+) by RT-PCR. We found an additional discrepancy in the CDI test results: EPEC O86 (RT-PCR eae+) was isolated from a fecal sample positive for GHA enzyme (CLIA) and negative for cdtA and cdtB (CLIA and PCR). However, the culture of *C. diff* was negative. These findings support the hypothesis that certain human bacterial pathogens that produce enterotoxins other than *C. diff*, as well as intestinal commensal microorganisms, including Klebsiella sp. and Proteus sp., contribute to false-positive EIA card tests for *C. diff* toxins A and B, which are the most widely used laboratory tests for CDI. **Conclusions**. CDI presents a significant challenge to clinical practice in terms of laboratory diagnostic management. It is recommended that toxin-only EIA tests should not be used as the sole diagnostic tool for CDI but should be limited to detecting toxins A and B. Accurate diagnosis of CDI requires a combination of laboratory diagnostic methods on which proper infection management depends.

## 1. Introduction

*Clostridioides difficile* (*C. diff*) is a Gram-positive, spore-forming, anaerobic, toxin-producing bacterium in the environment and the gastrointestinal tract of animals and humans. *C. diff* is a major cause of healthcare-associated infections and an increasing cause of morbidity and mortality among hospitalized adult patients [1,2]. Its prevalence is higher in the elderly population, as this group has a large number of risk factors, such as comorbidity, frequent exposure to healthcare, and greater consumption of antibiotics and antacids [2,3]. Furthermore, *C. diff* is a common cause of antibiotic-associated diarrhea (AAD), which results from disruption of the normal intestinal flora. AAD manifestations range from asymptomatic carriage, mild to fulminant diarrhea, toxic megacolon, and pseudomembranous colitis due to virulence factors: an enterotoxin (toxin A), which disrupts actin filament assembly, and a cytotoxin (toxin B), which mediates cell surface binding and intracellular translocation [4,5]. Spores and vegetative cells of *C. diff* colonize the human intestinal tract by the fecal–oral route; most vegetative cells are killed in the stomach, but spores of the toxin-producing bacillus are acid-resistant and germinate in the small intestine upon exposure to bile acids. Primary bile acids are known to stimulate this germination process. *C. diff* multiplies in the colon, where vegetative cells produce toxins A and B and hydrolytic enzymes [6,7,8]. The main reservoirs of *C. diff* include colonized or infected patients and contaminated environments and surfaces in hospitals, the hands of hospital staff, and long-term care facilities. *C. diff* can be shed from patients with CDI not only during the diarrheal period but also after the end of therapy [9,10,11,12,13]. Testing in this population should be performed when other infectious and noninfectious causes of diarrhea have been ruled out [9,10,11,12,13,14].

Accurate laboratory diagnosis of CDI is essential for effective prevention and treatment. Only a few diagnostic tests are routinely used, recommended by the European Society for Clinical Microbiology and Infectious Diseases (ESCMID) and the Infectious Diseases Society of America (IDSA). The enzyme immunoassays (EIAs) for *C. diff*. toxins A and B have a sensitivity of 75–85% and a specificity of 95–100% [5,9]. A glutamate dehydrogenase (GDH) assay detects the antigen in all *C. diff* strains. It is characterized by a rapid turnaround time and a specificity of almost 100%. However, it could not reliably distinguish whether the strain is toxigenic with a specificity of 59% [5,9,10]. Polymerase chain reaction (PCR) for detecting *C. diff* toxin genes is an additional, less frequently used test. The method has a sensitivity of 80–100% and a specificity of 87–99% [9,10,12]. Other less sensitive tests include stool culture or endoscopy to visualize pseudomembranous plaques in the colon and imaging studies, such as X-ray or computed tomography, to evaluate complications, such as toxic megacolon or perforation [6]. However, a single test cannot diagnose all cases, and laboratory diagnostic algorithms are often used [5,6,9]. This work aims to assess the risks of misdiagnosis, which are common in current routine laboratory diagnostic practices used in managing CDI in primary care settings, especially in Bulgaria [11].

## 2. Materials and Methods

### 2.1. Patient Selection

We defined patients as suspected of CDI in accordance with the ESCMID and IDSA recommendations that patients with unspecified and recent onset diarrhea characterized by the formation of ≥3 stools within 24 h should undergo testing for CDI. This study included 68 men and 48 women (12–80 years) admitted to two general hospitals. This study followed the Declaration of Helsinki and was approved by the Institutional Ethics Committee of “Ajibadem City Clinic-Tokuda” University General Hospital, Sofia, Bulgaria (Nr. 226/19 November 2024).

### 2.2. Samples

A total of 116 unformed fecal samples, defined as Bristol stool form scale type 6 and 7 [15], from inpatients with diarrhea, were examined before the etiological antibiotic exposure. The two hospitals are conventionally denoted by the following symbols: SF and DT. SF hospital provided data on 66 stool samples, positive for *C. diff* tested with PCR and chemiluminescence immunoassay (CLIA), whereas DT hospital provided data on 50 enzyme immunoassays (EIAs) positive for *C. diff* samples. The fecal samples were submitted to the National Reference Laboratory for Enteric Diseases on the day of deposition to ensure the efficacy of the bacterial culturing techniques. None of the source hospitals provide culture methods for *C. diff*.

EIA testing was conducted to determine the presence of toxin A and toxin B of *C. diff.* In the clinical laboratories or hospital departments, fecal samples were subjected to an EIA card test to detect both toxins A and B within two hours following the initial sampling. The result is conveyed within ten minutes of administering the card test drip.

Fecal samples were treated using the alcohol shock method to inhibit non-sporulating organisms and thereby improve the isolation of *C. diff*. A pea-sized portion of the sample was transferred to 350 µL of absolute alcohol. The suspension was allowed to stand at room temperature for 30 min, after which 100–150 µL of it was inoculated onto Brain Heart Agar, 10% sheep blood (Oxoid, UK), and incubated at 37–38 °C for 48–72 h under anaerobic conditions maintained with gas packs (Oxoid, UK). Furthermore, feces were cultured on MacConkey agar, Levin agar, and SS-agar at 37–38 °C for 18–20 h, and subsequently cultured after pre-enrichment in Selenite broth for 18–20 h at 37–38 °C.

The obtained cultures under anaerobic and aerobic conditions were identified using MALDI-TOF (Bruker, Bremen, Germany). On several occasions, API 20E biochemical tests were employed for final subspecies identification, followed by phenotypical serotyping (anti-*E.coli* sera, SSI, Hillerød, Denmark).

### 2.3. DNA-Extraction

Whole genomic DNA was extracted from each stool sample using a commercial DNA extraction kit (QIAamp PowerFecal Pro DNA Kit, Germany) following the manufacturer’s instructions. The extracted DNA samples were subsequently stored at −20 °C in preparation for subsequent molecular analysis.

### 2.4. Molecular Methods

A commercial gastrointestinal panel for detecting and differentiating bacterial, viral, and parasitic target pathogens (QIAstat-Dx Gastrointestinal Panel 2), including *C. diff* (toxin A/B). The following methods were used for differential diagnosis: the LAMP method for the detection of Salmonella spp. and simultaneous detection of *Campylobacter jejuni* and *C. coli* for 35 min; a commercial *E. coli* Typing RT-PCR detection kit (CerTest BIOTEC, Spain) for the detection of virulence genes of enterotoxigenic *E. coli*, enterohemorrhagic *E. coli*, Shiga toxin-producing *E. coli*, and enteropathogenic *E. coli* and enteroinvasive *E. coli* (ETEC, EHEC, STEC, EPEC, and EIEC). Additional molecular analysis by PCR was performed to investigate the presence of the LT-toxin-encoding *lth* gene in Enterobacteriales strains associated with diarrhea syndrome.

### 2.5. CLIA

Following the two-step diagnostic algorithm for CDI recommended by the ESCMID Clinical Guidelines (EU) and the Clinical Practice Guideline (USA), an automated CLIA assay was first performed to detect the GDH antigen, followed by a CLIA assay for the qualitative determination of toxins A and B in human feces (Diasorin’s LIAISON^®^, San Diego, CA, USA).

## 3. Results

Given the different financial resources available in the two hospitals, we assess the quality of *C. diff* diagnosis using the various methods used by the health units. Our work would help inform and develop appropriate policies for accurate laboratory diagnosis of CDI.

In this study, patients had experienced at least one hospitalization before the current admission. They were admitted to the health facility in the absence of diarrhea. A referral for examination for CDI was made based on the sudden onset of diarrhea (within 72 h after the hospitalization), as assessed using the Bristol Stool Form Scale (BSFS), types 6 and 7 [15], with a minimum of three bowel movements in 24 h. All patients had received antibiotics within the previous six months. The criteria mentioned above collectively defined the profile of the subjects as susceptible to CDI. A total of 116 patients (68 men and 48 women), aged between 12 and 80 years, were included in this study.

Forty toxigenic *C. diff* strains (40/116, 34.4%) were isolated from all diarrheal stool specimens tested, of which 38/40 produced toxin B, and 2/40 were positive for both toxins A and B. Toxigenic profiles were determined by the EIA card test for free toxins in feces in DT hospital and from the bacterial culture at National Centre of Infectious and Parasitic Diseases (NCIPD), as well as the CLIA assay (Diasorin’s LIAISON^®^) for the detection of *C. diff* GDH antigen, followed by an assay for the qualitative determination of toxins A and B in human feces. Additionally, the samples that were identified as GDH+, cdtA−, and cdtB− or those that were identified as GDH−, cdtA+, and/or cdtB+, were subjected to further analysis via PCR for *C. diff* toxins A and B (QIAstat-Dx) at ST hospital. The culture of the *C. diff* bacillus was obtained and identified with MALDI- TOF at NCIPD (Figure 1).

Of the 50 fecal samples (50/116) tested with EIA assays (NADAL^®^) and were found positive for *cdtA*+ (6/50) and *cdtB*+ (44/50), 17 stool cultures of *C. diff* (17/50) were isolated, all of which were positive for the *cdtB*. The examination of these 4/50 fecal samples yielded positive results for detecting free *cdtA*+ and *cdtB*− (by EIA), demonstrating the presence of diarrheal agents through aerobic and anaerobic stool culturing. The identified agents were *C. perfringens* (2/6), monocultures of two hyper-mycotic *Klebsiella pneumoniae* (2/16), and two *Proteus mirabilis* strains (2/6). Furthermore, the presence of the LT-toxin-encoding *lth* gene was investigated in the Klebsiella and Proteus strains using PCR. The specific primers were designed based on the *E. coli* gene from the GENEBANK database, and the PCR mix and reaction conditions were previously described by Janczura et al. [16]. The analysis of *cdtA*−/*cdtB*+ (by EIA) stool samples that were culture-negative for *C. diff* (31/48) evidenced the presence of pathogenic agents. This was conducted through aerobic stool culturing and LAMP molecular methodology as a differential diagnostic approach. A single fecal sample was identified as positive for Salmonella sp. (1/10) using LAMP, while two additional samples were found positive for *C. jejuni* (2/10). Bacterial cultures were obtained from these samples and subsequently confirmed by MALDI-TOF and biochemical typing of *Salmonella* as *enterica* subsp. *arizonae* with an API 20E test. Three out of ten fecal samples yielded enteropathogenic *E. coli* (EPEC), which were phenotypically serotyped (anti-*E. coli* sera, SSI, Denmark) as enterotoxigenic *E. coli* (ETEC) O142 and two strains of Shiga-toxin producing *E. coli* (STEC) O145 and STEC O138. Following molecular genotyping with the *E. coli* typing RT-PCR kit (CerTest) the bacterial cultures with detected virulence genes were tested for heat-labile enterotoxin (*lt*) and heat-stable enterotoxin (st) in *E. coli* O142, as well as genes for Shiga toxins (*stx1+* and *stx2+*) in O145 and *stx1*+ in STEC O138 (Table 1).

All 66 fecal samples from the SF hospital were subjected to primary testing for CDI through a commercial enzyme-linked immunosorbent assay (ELISA) for detecting the GDH enzyme, followed by quantification of the *cdtA/B* genes. If the fecal specimen was not submitted to the clinical microbiology laboratory for testing within 2–3 h following disposal, even without a positive GDH result, toxin testing was conducted. For samples that tested positive for GDH and *cdtA*−/*cdtB*− and for samples that tested negative for GDH but positive for toxins, a repeat confirmatory test utilizing PCR for *cdtA/B* was performed.

Two of the 66 cdtB+ samples exhibited a similar misdiagnosis when analyzed using CLIA. However, the PCR analysis indicated that they were *cdtB*-negative. In contrast, the LAMP method identified a positive result for *C. jejuni* in one sample, and another was STEC positive (stx1+/stx2+) by RT-PCR. Furthermore, there was an additional discrepancy in the CDI testing results—EPEC O86 (RT-PCR eae+) was isolated from one fecal sample (1/66) that was positive for GDH enzyme (CLIA) and *cdtA*− and *cdtB*− (CLIA and PCR assays). However, the *C. diff* culture was negative (Table 2) (Figure 2).

## 4. Discussion

To realistically assess the management of the diagnostic process of CDI in health facilities, we selected one of the largest hospitals in the capital and a randomly selected smaller hospital in the province, both of which regularly register cases of acquired diarrhea after admission and healthcare, serving a large number of patients from the country. In this study, the methods used to diagnose CDI were available only in both health facilities. EIA tests were mainly used, which were limited to detecting *cdtA* and *cdtB* and could not detect the GDH enzyme. In addition, there was a rarely used automated CLIA assay for the detection of *C. diff* GDH antigen and an assay for the qualitative determination of *cdtA* and *cdtB* in human feces. At the Enteric Infections Unit, Department of Microbiology, NCIPD, we performed a battery of tests, including culture of fecal specimens and molecular, biochemical, and phenotypic methods for identifying and serotyping enteric pathogens. The accurate laboratory diagnosis of CDI requires a combination of diagnostic methods and clinical management practices. Interpreting the results requires considering the patient’s symptoms, such as any new onset of diarrhea, defined as three or more loose stools in 24 h. It is possible that the symptoms are due to an active CDI. In addition, a positive result from a complex of laboratory tests, in the absence of clinical symptoms, may indicate colonization rather than active infection [8,10,17,18]. The patient’s history, previous CDI, antibiotic exposure, and comorbidities should also be considered when interpreting the results. In this study, patients had at least one hospitalization prior to the current admission. Patients were admitted to the health facility in the absence of diarrhea. Laboratory tests to diagnose CDI identified toxin B as the predominant toxigenic profile of *C. diff* isolates from hospital patients. This is consistent with findings from other national studies in Bulgaria, which concluded that most pathogenic Bulgarian *C. diff* isolates contain the tcdB+ gene [19]. Furthermore, according to our available sources, laboratory-confirmed cases of *C. diff* expressing both toxins simultaneously appear to remain very rare in the country [19,20]. It is suggested that a significant number of EIA false-positive rapid tests for *C. diff* toxA+ (12/50; 12%) show cross-reactions with enterotoxins produced by members of the Enterobacteriales, namely *Klebsiella pneumoniae* and *Proteus mirabilis* [13,16]. These cross-reactions are likely to be observed in cases of antibiotic-associated and sporadic diarrhea, as well as in cases of *Clostridium perfringens*, which is recognized as the cause of foodborne outbreaks [21]. In addition, Forward et al. have raised the prospect of identifying toxins produced by *C. diff* and *C. perfringens* in cases of community-acquired diarrhea using commercially available EIAs [21,22,23]. Laboratory tests such as EIAs for *cdtA* and *cdtB* are of great importance in diagnosing infection. However, they confirm the presence of the bacteria and their toxins only if they are performed according to two-step diagnostic protocols. First, to detect the enzyme GDH, which confirms the presence of *C. diff*, then proceed to demonstrate the presence of toxins A and B [9,10]. Although EIA tests that combine GDH with ToxA/B are commercially available, Bulgarian microbiology laboratories use binary *C. diff* toxin tests in hospital practice for CDI management [19,20]. The reason is most likely the lower the cost of the tests. Their widespread use carries significant risks for both misdiagnosis and inappropriate treatment and the possible occurrence of nosocomial infection. Conversely, many EIA and CLIA tests report *cdtB+* (7.7%), although they lag behind campylobacteriosis, salmonellosis, and STEC infections, which could be key markers for CDI superdiagnosis. In the context of CDI prevention, clinicians typically order only those tests that are available to them. It can be assumed that toxin-positive tests without GDH may serve as a key marker for non-clostridial bacterial enterocolitis. However, a comprehensive study including a significant number of patients with diarrhea and a set of confirmatory tests is needed to answer this question. Alternatively, a full laboratory differential diagnosis of the syndrome should be performed, supported by clinical data. However, in clinical practice, rapid assays, ELISA, PCR, and CLIA are often used as stand-alone tests rather than as components of an algorithm, including stool culture, which is considered the gold standard. This, in turn, can lead to misclassification of *C. diff* colonization as CDI [12,18,20]. To define an infection as healthcare-associated, a number of specific circumstances must be taken into account. First, if symptoms began 72 h or more after admission, and second, if an individual is diagnosed with CDI within four weeks of discharge from any healthcare facility. Finally, in cases where the infected individual was a healthcare worker in contact with patients [23,24]. Gateau et al. also concluded that none of the stand-alone tests combine high sensitivity and specificity for diagnosing CDI. Definitive diagnosis can be achieved by applying a two- or three-step algorithm. The algorithms are set out in ESCMID and include a highly sensitive screening test verified by a more specific test to detect free toxins. Nucleic acid amplification tests can detect asymptomatic carriers of a toxigenic strain [25]. Ziaei et al. also described the shortcomings of cheap and fast conventional methods. They proposed combining modern methods for diagnosing *C. diff* to distinguish between carriage and active infection [26]. Applying these sophisticated laboratory methods facilitates the identification of cases of healthcare-associated CDI. This leads to appropriate control and prevention measures. Finally, it is important to consider the potential for pathogen misclassification and inappropriate antimicrobial therapy when using a rapid toxin test without culture to diagnose and manage CDI. Therefore, the implementation of appropriate etiological therapy is recommended to avoid the risk of recurrent CDI, as well as those associated with antibiotic polypharmacy. This is the primary and safest way to limit the spread of healthcare-associated CDI.

Our work is limited by insufficient data from the official databases and registers in Bulgaria. This does not allow us to conduct a broad retrospective description and form a cohort design for the country at the moment.

## 5. Conclusions

A comprehensive approach to the diagnosis of CDI is essential to ensure accurate diagnosis and appropriate treatment. This requires considering a range of factors, including clinical presentation, antibiotic exposure, history of hospitalizations, previous CDI, and presence of colonoscopy-proven pseudomembranous colitis. The only reliable approach to etiological diagnosis is using a combination of laboratory methods. For example, IDSA recommends several multi-step options. The most suitable for the conditions in Bulgaria is the detection of GDH and EIA for both toxins. At the same time, molecular methods require specific equipment, and their routine use is not appropriate, although it has its place in the case of diagnostic difficulties. In cases of treatment failure, antimicrobial susceptibility is a necessary method. We strongly reject using the one-step EIA method for detecting both toxins as an unreliable diagnostic tool.

## Figures and Tables

**Figure 1 diagnostics-15-01166-f001:**
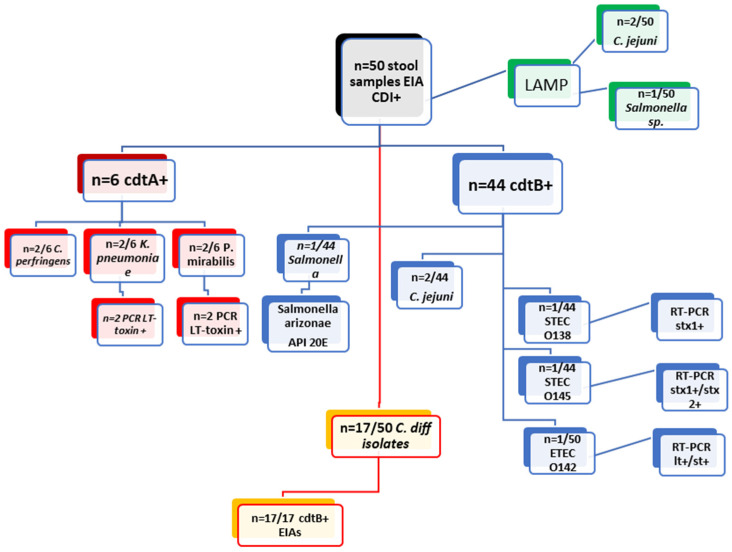
Examination of 50 clinical stool samples cdtA+/cdtB+ by EIAs from DT hospital. Stool culturing, RT-PCR for *E. coli* typing, LAMP assay for Salmonella and Campylobacter, and EIAs cdtA/B.

**Figure 2 diagnostics-15-01166-f002:**
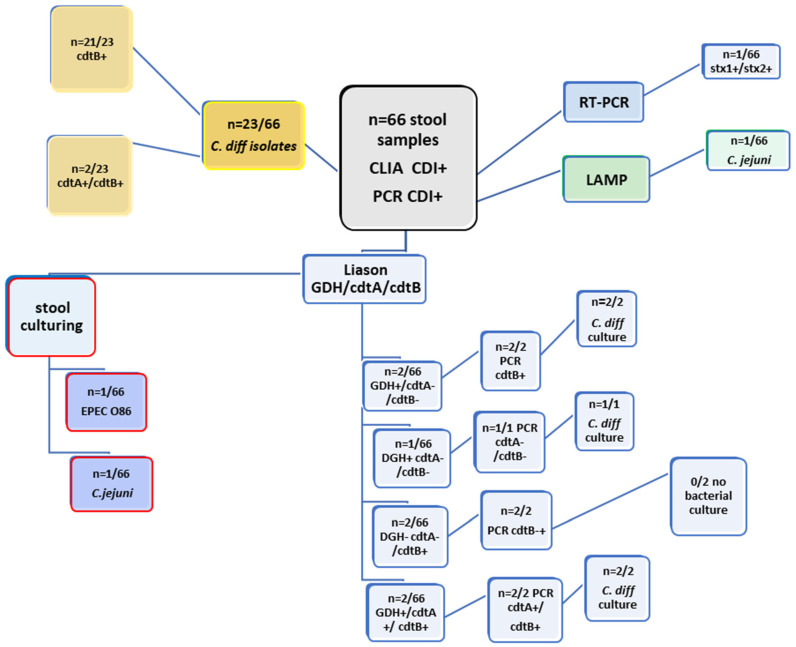
Examination of 66 clinical stool samples CLIA and RCR tested for CDI from SF hospital. The differences in the laboratory diagnostic methods are presented here. Stool culturing, RT-PCR for *E. coli* typing, LAMP assay for Salmonella and Campylobacter, and EIAs for *cdtA/B*.

**Table 1 diagnostics-15-01166-t001:** Results from examining 50 stool samples from hospitalized patients from DT hospital by various laboratory methods—stool culturing, RT-PCR for *E. coli* typing, LAMP assay for Salmonella and Campylobacter, and EIAs *cdtA/B*.

Examination of 50 Clinical Stool Samples cdtA+/cdtB+ by EIAs from DT-Hospital									
Methods	EIA cdtA+ (*n* = 6/50)			EIA cdtB+ (*n* = 44/50)					
culture	*C. perfringens (2/6)*	*K. pneumoniae (2/6)*	*P. Mirabilis (2/6)*	*Salmonella*	*C. difficile* (17/44)	*C. jejuni* (2/44)	STEC O138 (1/44)	STEC O145 (*n* = 1)	ETEC 142 (1/44)
PCR	x	LT toxin (*n* = 2)	LT toxin (*n* = 2)	x	x	x	stx1+ (*n* = 1)	stx1+/stx2+ (*n* = 1)	lt+/st+ (*n* = 1)
API20E	x	x	x	*Salmonella arizonae*	x	x	x	x	x
LAMP (*Salmonella/Campylobacter*)	negative	negative	negative	*Salmonella*	negative	*C. jejuni* (2/44)	negative	negative	negative

**Table 2 diagnostics-15-01166-t002:** Results from examining 66 stool samples from hospitalized patients from SF hospital by various laboratory methods—CLIA and RCR assays for CDI stool culturing, RT-PCR for *E. coli* typing, LAMP assay for Salmonella and Campylobacter, and EIAs for *cdtA/B*.

Examination of 66 Clinical Stool Samples by CLIA (GDH; cdtA/B) and PCR (cdtA; cdtB) from SF Hospital								
**Methods**	Liaison GDH	Liaison cdtA	Liaison cdtB	PCR cdtA	PCR cdtB	Culture	LAMP	RT-PCR (STEC)
	2/66	negative	negative	negative	2/66	*C. difficile* (*n* = 2/66)	negative	x
	1/66	negative	negative	negative	negative	*C. difficile* (*n* = 1/66)	negative	x
	negative	negative	2/66	negative	2/66	negative	negative	x
	negative	negative	1/66	negative	1/66	*C. jejuni* (*n* = 1/66)	*C. jejuni*	x
	negative	negative	1/66	negative	1/66	EPEC O86	negative	stx1+/stx2+
	2/66	2/66	2/66	2/66	2/66	2/66	negative	x
	negative	negative	21/23	negative	21/23	*C. difficile* (*n* = 23/66)	negative	x
	negative	2/23	2/23	negative	2/23	*C. difficile* (*n* = 23/66)	negative	x

## Data Availability

Data generated or analyzed during this study are included in this manuscript.

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
