# Peer review of "Potential for Misinterpretation in the Laboratory Diagnosis of Clostridioides difficile Infections"

_diagnostics, 2025, doi:10.3390/diagnostics15091166_

Round 1
Reviewer 1 Report
Comments and Suggestions for Authors
I commend the authors for their comprehensive work, which addresses the complexities associated with both the clinical and laboratory diagnosis of Clostridium difficile (C. diff). Below are my detailed comments and suggestions for improving the manuscript:
-
Introduction (Epidemiology Section): I recommend removing the section in the introduction that discusses the epidemiology and clinical features of C. diff infection in children. This content does not appear relevant to the study's results and may detract from the focus of the manuscript.
-
Introduction (Diagnostic Methods): The introduction should include well-established (or at least estimated) sensitivity and specificity values for the diagnostic methods discussed. This information would provide important context for the study.
-
Introduction (Conciseness and Guidelines): The introduction should be succinct and focused, incorporating a summary of the standard diagnostic algorithms recommended by ESCMID (European Society of Clinical Microbiology and Infectious Diseases) and the CDC (Centers for Disease Control and Prevention). This would provide clear grounding for the readers.
-
Patient Data Placement: Patient-related information, such as demographics and medical background, is currently placed in the discussion section. This data should be relocated to the results section in order to adhere to conventional scientific reporting structures.
-
Methods (Ethics Statement): The methods section should explicitly state that Institutional Review Board (IRB) approval was obtained for the study. This is an essential inclusion for ethical considerations and compliance with reporting standards.
-
Speculation in Discussion: The discussion should avoid unsupported or speculative statements, such as those found in lines 291–296. Ensuring the discussion is grounded in the data generated by the study will maintain scientific rigour.
-
Comparative Analysis: The discussion would benefit from comparisons with findings from similar studies conducted outside of Bulgaria. Providing a broader context for the results would strengthen the manuscript and allow for a more meaningful interpretation of the data.
-
Relevance of Community vs. Hospital Infections: The discussion concludes with an extensive analysis of the differences between hospital-acquired and community-acquired infections. However, the relevance of this discussion to the current study’s scope is unclear. If unrelated, this section should be revised or removed.
-
Title Revision: The current title does not adequately reflect the primary focus of the study. As the research does not centre on patients’ risk factors for C. diff acquisition but rather evaluates the diagnostic accuracy of various laboratory tests, the title should be revised. Consider including reference to the retrospective, cohort design, as well as the geographical context of the study, i.e., Bulgaria.
Comments on the Quality of English Language
The quality of the English language requires significant improvement. The current manuscript lacks linguistic coherence, with frequent instances of overly complex sentence structures that impede readability and flow. Additionally, there is inconsistency in the use of UK and American English spelling. It is recommended that a single style, either UK or American English, be selected and implemented consistently throughout the text to ensure clarity and uniformity.
Author Response
The reviewers' responses are in the attached file.

Reviewer 2 Report
Comments and Suggestions for Authors
The author should change the title and background of the study or the Methods and the results.
The term ‘risk factor’ as the author stated, for the objective of the study is not appropriate. The term ‘risk factor’ is associated with the occurrence of disease. When read to the results, it implies that specificity and sensitivity of toxigenic culture is somehow low. There are other variables that interfere with the test results.
Abstract
There is incongruent writing. The author writes about Clostridium difficile risk factors in the background; the result shows false positive and false negative of diagnostic test; and the author concludes that several diagnostic test to confirm C. difficile diagnosis.
Introduction and Methods
The author should have the 116-patient characteristic baseline from the two hospitals.
- difficile risk factors: Page 2 lines 78-82: Antibiotic exposure (especially fluoroquinolones, third or fourth-generation cephalosporins, clindamycin, carbapenems), prior hospitalization, advanced age, an increase in the severity of underlying illness, immunocompromising conditions, gastrointestinal manipulation, and proton-pump inhibitors are all risk factors for C. difficile infection.
- difficile diagnostic tests: Page 3 lines 113: section 2.2. Samples, 2.3. DNA-Extraction, 2.4. Molecular Methods, 2.5. Chemiluminescence Immunoassay (CLIA)
The author should explain the association the diagnostic test in the Introduction (Page 3 lines 95-101) and in the Methods section (Page 3, 4).
Discussions and Conclusions
The discussion and conclusions should follow the author decision about the objective of the study and the results of the study.
Comments on the Quality of English LanguageModerate
Author Response

(The authors gave the same response as above.)

Round 2
Reviewer 1 Report
Comments and Suggestions for Authors
The revised version is indeed far better then the original one, it now has a logical and scientific flow.
The conclusion requires amendment, as conducting all tests, including culture and sensitivities, is not a pre-requisite to making a clinical diagnosis of C. diff according to any guideline.
Unfortunately, I was not provided with the cover letter which surely would have shed some more light on the changes made.
Comments on the Quality of English LanguageEnglish language still is in dire need of improving (e.g., patients conducted in the study?; spelling is still a mix of American and UK English e.g., hospitalizations and hospitalisaitons)
Author Response
The answers are in the attached file.

Reviewer 2 Report
Comments and Suggestions for Authors
What is 'CDI superdiagnosis' (page 12 line 339)?
Does the author always suggest performing a culture and toxin test for CDI diagnosis (as stated in the conclusion section)? Recent guideline 'or' aside 'and' (Cymbal, 2024).
Reference:
-
Management of Clostridioides difficile Infection: Diagnosis, Treatment, and Future Perspectives
Cymbal, Michael et al. The American Journal of Medicine, Volume 137, Issue 7, 571 - 576
Moderate
Author Response
The answers are in the attached file.
